# Siam Deep Feature KCF Method and Experimental Study for Pedestrian Tracking

**DOI:** 10.3390/s23010482

**Published:** 2023-01-02

**Authors:** Di Tang, Weijie Jin, Dawei Liu, Jingqi Che, Yin Yang

**Affiliations:** 1College of Mechanical Engineering, Zhejiang University of Technology, Hangzhou 310014, China; 2China Aerodynamics Research and Development Center, High Speed Aerodynamic Institute, Mianyang 621000, China

**Keywords:** pedestrian tracking, machine learning, YOLO, Siamese CNN, ROS

## Abstract

The tracking of a particular pedestrian is an important issue in computer vision to guarantee societal safety. Due to the limited computing performances of unmanned aerial vehicle (UAV) systems, the Correlation Filter (CF) algorithm has been widely used to perform the task of tracking. However, it has a fixed template size and cannot effectively solve the occlusion problem. Thus, a tracking-by-detection framework was designed in the current research. A lightweight YOLOv3-based (You Only Look Once version 3) mode which had Efficient Channel Attention (ECA) was integrated into the CF algorithm to provide deep features. In addition, a lightweight Siamese CNN with Cross Stage Partial (CSP) provided the representations of features learned from massive face images, allowing the target similarity in data association to be guaranteed. As a result, a Deep Feature Kernelized Correlation Filters method coupled with Siamese-CSP(Siam-DFKCF) was established to increase the tracking robustness. From the experimental results, it can be concluded that the anti-occlusion and re-tracking performance of the proposed method was increased. The tracking accuracy Distance Precision (DP) and Overlap Precision (OP) had been increased to 0.934 and 0.909 respectively in our test data.

## 1. Introduction

Since the 21st century, public security problems encountered by various countries have been increasing. With the rapid development of technology, public security crises have become more hidden [1]. Fortunately, the invention of computer vision has changed many traditional fields of human technological activities [2,3,4], especially in some security fields, such as public security [5,6]. Installing surveillance cameras in dense venues is an effective way to reduce crimes and ensure social security [7]. To reduce the number of fixed cameras and continuously supervise moving pedestrians’ misbehaviors, an airborne unmanned aerial vehicles (UAV) system with pedestrian tracking came to be used in real-time tracking. Unlike the tracking method in traditional surveillance cameras, the UAV pedestrian tracking system requires a more lightweight network with fast-running abilities.

Most of the state-of-art pedestrian tracking algorithms usually use the tracking-by-detection framework. The framework can be considered to be an estimation problem, composed of pedestrian detection and data association [8]. In the former, when a video sequence is obtained, we need to detect the frame of the object by frame for the association. In the latter, we link together the detection object in different frames by using data association strategies, which depend on features such as appearance. Yu et al. [9] showed that in the framework of tracking-by-detection, detection quality could seriously affect the performance of pedestrian tracking. Therefore, to improve the performance of pedestrian tracking, an accurate detector is necessary. Most of the traditional pedestrian detection algorithms [10,11] mostly utilize the method of sliding windows to traverse the entire image to locate the positions of objects and then extract robust features of the image, such as Scale Invariant Feature Transform (SIFT), Histogram of Gradient (HOG), and so on. After that, the features are sent into classifiers with the traditional machine learning method Adaptive Boost (Adaboost) [12] or Support Vector Machines (SVM) [13]. However, these algorithms rely on manually designed classification features, and these manually designed features are not suitable for multi-scale detection in complex backgrounds. In recent years, with the rapid development of Convolutional Neural Networks (CNNs) which can reduce the image to a form that can be easily processed without losing vital features for prediction, image-processing methods based on CNN have been widely used in the field of object detection and classification [14,15,16]. Since then, the deep detection mode has been widely developed. It can be divided into two categories: one-stage networks and two-stage networks [17,18]. Regions with CNN feature (RCNN) [19] is a typical two-stage method of object detection which proposes a selective search method to find candidate boxes. The Fast-RCNN method [20] and Faster-RCNN [21] were proposed by Ross Girshick to further improve performance, and it was also suggested that using EdgeBoxes instead of the selective search method would reduce the time for candidate box proposals. However, the detection speed of two-stage networks was far inferior to that of one-stage networks. Nowadays, YOLO [22] is the most popular one-stage object detection method which treats object detection as a regression task. YOLOv3 [23] predicts the coordinates of the possible bounding box, object class, and confidence of object class. It has been widely used in vehicle [24], pedestrian [25,26], fire [27], and even medical cell detections [28,29], etc. However, the huge computation cost of YOLOv3 makes it difficult to achieve real-time detection in small airborne UAV systems. Therefore, the allocation of computing resources in YOLOv3 is an alternative and efficient way to improve the computational efficiency.

The real-time deep tracking method has been widely used in many areas; however, its application in UAV system detections is challenging because of its limited computations. Fortunately, Correlation Filter (CF)-based tracking methods have attracted increasing interest due to their high computational speed and high operating efficiency. Dense sampling in these methods is approximated by generating a circulant matrix. Each row in the circulant matrix denotes a vectorized sample. With such kind of representation, a regression model can be efficiently solved in the Fourier domain. It is a method to obtain more realistic and reliable regression coefficients at the expense of losing some information and reducing little accuracy. Bolme et al. [30] presented a new type of correlation filter, a Minimum Output Sum of Squared Error (MOSSE) filter, which produced stable correlation filters when initialized using a single frame. Subsequently, Henriques et al. proposed the Circulant Structure of Tracking-by-detection with Kernels (CSK) [31] which provided a link to Fourier analysis that opens up the possibility of extremely fast learning and detection with the Fast Fourier Transform. Immediately, the authors derived a new kernelized correlation filter (KCF) [32] that unlike other kernel algorithms had the same complexity as its linear counterpart. However, the KCF is a method to obtain faster computation speed by reducing accuracy. It has a fixed template size resulting in the occlusion problem. Furthermore, the HOG feature used in KCF leads the tracking method to easily lose its target. Considering these cases, a Siamese Convolutional Neural Network [33] (Siamese CNN) is used in the proposed method. It consists of two identical artificial convolutional neural networks each capable of learning the hidden representation of an input vector, work parallelly in tandem and comparing their outputs at the end, usually through a cosine distance. That is the similarity between the two images.

In all literature, few studies have been conducted to design a lightweight network with fast-running abilities to track a moving body even though it may be occluded. Thus, the main topics of this paper are to propose a Siam Deep Feature KCF method, which has high computational efficiency for pedestrian tracking. Dummy tracking experiments were carefully designed and performed to study the influences of moving speed, rotating speed, and occlusion on tracking accuracy and efficiency. Thereafter, pedestrian tracking experiments were performed to verify its performance in real-time tracking. The proposed method has been irrefutably validated and it is considered to be suitable for unmanned aerial vehicles system due to its lightweight features. The structure of this paper is described as follows: Section 2.2 introduces the object detection algorithm, Section 2.3 describes the data association method, and in Section 3 and Section 4, we present the experimental results and give the conclusion.

## 2. Materials and Methods

### 2.1. DFKCF Method Coupled with the SiamCSP

To port the pedestrian tracking method into the airborne UAV systems, a Siam-DFKCF method is proposed. In this method, the frame topic is published by the *Usb_Cam* node, which mainly publishes image frames (/img_frame) and time stamps (/time_stamp) to other nodes. A lightweight YOLOv3 node *YOLO-ECA* subscribes to the image topic (/img_frame) and time stamps topic (/time_stamp). Therefore, the pedestrian can be detected followed by the publishing of the Roi topic (/roi_initial). Thereafter, the tracking node *KCF* subscribes to the Roi topic, tracks the pedestrian, predicts the pedestrian location, and publishes the relevant image topic (/roi_predict). After that, a lightweight Siamese CNN node *SiamCSP* subscribes to both the Roi and image topic, then calculates the image similarity (/similarity) between the previous two topics. During the tracking, a Motor drive node *Gimbal_Motor* is used to perform the tracking command. The ROS Master (Robot Operating System [34]) registers the published topics of each node followed by a subscription from the Master, as shown in Figure 1.

The process of this Siam-DFKCF algorithm includes four main stages: Feature extraction, Feature tracking, Feature re-extraction, and Execution stage.

(1)Feature extraction: The *Usb_Cam* node publishes the frame to the *YOLO-ECA* node, and the *YOLO-ECA* node detects and extracts the received frame to obtain the feature of the target (/roi_initial), which is received by the *KCF* node and the *SiamCSP* node.(2)Feature tracking: The tracking box (/roi_predict), including both size and the relative location, is predicted by the *KCF* node through correlation filter processing. The topic about the location is published to the *Gimbal_Motor* node so that the camera can track the pedestrian in real time. Thereafter, the received feature (/roi_initial) is compared with the predicted result (/roi_predict) by the *SiamCSP* node to obtain the similarity (/similarity).(3)Feature re-extraction: When the image similarity (/similarity) is below the minimum threshold, go to step (1) to recalculate the correlation feature model in the *KCF* node. Otherwise, go to the next step.(4)Execution stage: The relative position of (/roi_predict) in the previous frame and the current frame is calculated and then transferred to the *Gimbal_Motor* node to perform tracking.

The schematic diagram of the Siam-DFKCF algorithm is shown in Figure 2.

### 2.2. An Improved YOLOv3 Based on Efficient Channel Attention

YOLOv3 [23] is a one-stage network based on a regression method which extracts the features and directly predicts and classifies input images. It does not need to generate a large number of candidate windows compared with two-stage networks, and has excellent recognition speed and detection accuracy. Specifically, YOLO-v3 first resizes the input image to a 416 × 416 pixel image and then feeds it into deep neural networks for training in the Traditional YOLOv3. Then the 416 × 416 pixel image is divided into S × S grids, and each grid is responsible for predicting the pedestrian within the image. In addition, 26 × 26 feature maps are usually fused with 52 × 52 feature maps via up-sampling. Similarly, the 13 × 13 feature maps are fused with the 26 × 26 maps via up-sampling in the traditional multi-scale feature fusion. This multi-scale feature fusion and Darknet-53 [35] constitute the feature extraction network of YOLO-v3. Detection accuracy can be greatly improved by the combination of the deep feature and the shallow feature. However, it still has several challenges when applied towards pedestrian detection. Therefore, the Feature Extraction Network, Feature Pyramid Network (FPN), and Detection Network are combined to establish an Efficient Channel Attention YOLOv3method (ECA-YOLO), which is shown in Figure 3.

(1)Feature Extraction Network:

In Figure 3, the backbone feature extraction network reshapes the input image into a 608 × 608 × 3 RGB image which is then fed to the CBM. The CBM represents a complete convolutional layer, including three operations: Convolution Operation (Conv), Batch Normalization (BN), and Mish activation function. The identified 128 × 152 × 152 feature is then fed to the SCSPBody feature extraction network, as shown in Figure 4. To further improve the accuracy and efficiency of the feature extraction network, SqueezeNet [36] and Cross Stage Partial Network (CSPNet) are introduced into the backbone of the Squeeze Cross Stage Partial Body (SCSPBody). Where the CSPNet is the backbone network of YOLO-v4 [37] with an enhancement of the learning capacities of CNN, the SuqeezeNet is the smaller CNN architecture that requires less communication across servers during distributed training with equivalent accuracy. Furthermore, it requires less bandwidth to export a new model from the cloud to an autonomous car or UAV and is more feasible to deploy on FPGAs and other hardware with limited memory.

SCSPBody divides the feature maps of the input feature layer into two parts and then concatenates them through the cross-stage hierarchical structure. Specifically, SCSPBody1 is composed of two Conv 3 × 3, one Conv 1 × 1, four residual units, and four fire units, as shown in Figure 4a. Firstly, the input feature layer with a size of 128 × 152 × 152 is convoluted to Conv 256 × 3 × 3/2 to obtain the feature maps with a size of 256 × 76 × 76. Secondly, the network is divided into two parts: one part uses Conv 128 × 3 × 3 to generate feature maps with a size of 128 × 76 × 76, and the other part uses four residual units for feature extraction. The number of channels is adjusted to 128 by Conv 128 × 1 × 1, and then four fire units are used to continue extracting features to obtain 128 × 76 × 76 feature maps. Conv-BN-LeakReLU 16 × 1 × 1, Conv-BN-LeakReLU 64 × 1 × 1, Conv-BN-LeakReLU 64 × 3 × 3 are applied to each fire unit. Finally, the 128 × 76 × 76 feature maps of the two parts are concatenated as the output layer (256 × 76 × 76) using Equation (1)
(1)$$x_{𝓁}=H\left([x_{fire},x_{Conv}\right]),$$
where $x_{fire}$ is the output feature of Conv 128 × 1 × 1 and $x_{Conv}$ is the output feature of the fire unit. Similarly, SCSPBody2 is composed of two Conv 3 × 3, one Conv 1 × 1, and eight fire units as shown in Figure 4b. Firstly, the input feature layer is convoluted to Conv 256 × 76 × 76 to obtain the 512 × 38 × 38 feature maps. Secondly, the network is divided into two parts: one part uses Conv 256 × 3 × 3 to generate 256 × 38 × 38 feature maps, and the other part uses Conv 128 × 1 × 1 to adjust the number of channels. Then four fire units are used to extract features to obtain 128 × 38 × 38 feature maps, and then further used to extract features to obtain 256 × 38 × 38 feature maps. Conv-BN-LeakReLU 16 × 1 × 1, Conv-BN-LeakReLU 64 × 1 × 1 and Conv-BN-LeakReLU 64 × 3 × 3 are applied to the first four-unit; Conv-BN-LeakReLU 32 × 1 × 1, Conv-BN-LeakReLU 128 × 1 × 1, and Conv-BN-LeakReLU 128 × 3 × 3 are applied to the second four-unit. Finally, the 256 × 38 × 38 feature maps of the two parts are concatenated as the 512 × 38 × 38 output layer. In residual units, the Conv 64 × 1 × 1 layer compresses the number of channels for the feature layer ι, and then the Conv 128 × 3 × 3 is used to enhance feature extraction and expand the number of channels. The feature layers $x_{\iota -1}$ and $H\left(x_{\iota -1}\right)$ are connected by a shortcut. Finally, $x_{\iota }$ is defined in Equation (2) [38].
(2)$$x_{\iota }=H\left(x_{\iota -1}\right)+x_{\iota -1},$$

The fire units reduce the amount of computation during model training and reduce the size of the model file, which is more convenient for model saving and transmission. The operation of the Squeeze channel $S_{1\times 1}$ and the Expand channel $E_{1\times 1}$, $E_{3\times 3}$ are defined in Equation (3) [39].
(3)$$S_{1\times 1}=\frac{E_{1\times 1}}{4}=\frac{E_{3\times 3}}{4},$$

The Feature Extraction Network of the YOLO-ECA model is listed in Table 1.

(2)Feature Pyramid Network (FPN)

In deep convolutional neural networks, the low-level (high-resolution) feature layer contains more detailed information, and the high-level (low-resolution) feature layer contains more semantic information. As the network layer gradually deepens, the detailed information continues to decrease, while the semantic information continues to increase. To achieve multi-scale object detection, the feature pyramid network fuses high-level semantic information with low-level detailed information of different layers, which can improve feature extraction capabilities and the detection accuracy of small objects.

To obtain multi-scale semantic information about a pedestrian, motivated by the works of [40,41,42], a structure of the feature pyramid network is adopted in this paper, as shown in Figure 5. The multi-scale prediction process is as follows. Firstly, the large feature layer (LFL0), medium feature layer (MFL0), and small feature layer (SFL0) are effective feature layers extracted from the backbone network of the YOLO-ECA model. Secondly, the feature layer SFL-ECA1 is obtained after the Efficient Channel Attention (ECA) operation of feature layer SFL0, and then feature layer SFL-ECA1 is fused with feature layer MFL0 via up-sampling to generate feature layer MFL1. The feature MFL-ECA1 is obtained after the ECA operation, then MFL-ECA1 is fused with feature layer LFL0 via up-sampling to obtain feature layer LFL1. Finally, feature layer LFL-ECA1 is obtained after the Efficient Channel Attention operation of feature layer LFL1, and then the feature layer LFL-ECA1 is fused with feature layer MFL1 via down-sampling to generate feature layer MFL2. The feature layer MFL-ECA2 is obtained after the ECA operation of feature layer MFL2, and then the feature layer LFL-ECA2 is fused with feature layer SFL-ECA1 via down-sampling to obtain feature layer SFL2. The feature layers LFL1, MFL2, and SFL2 are connected to three CBL units for multi-scale prediction, respectively. Feature reuse is further realized by the top-down and bottom-up feature fusion strategies, which can effectively improve the prediction accuracy of a pedestrian.

Motivated by the works of ECA Net [43], adaptive attention is integrated into the FPN network using the Efficient Channel Attention structure which is shown in Figure 6. Firstly, the feature layer (FL0) is extracted from the backbone network of the YOLO-ECA or fused by FPN. Secondly, the feature layer FL-GAP0 is obtained after the global average pooling (GAP) operation of feature layer FL0, and then the Efficient Channel Attention weight FL-Weight is obtained by Conv1D and the Sigmoid activating. Finally, feature layer FL-ECA0 is calculated by multiplying FL0 by FL-Weight.

(3)Detection Network

The prediction box (/roi_predict) is calculated using the operations of the CBL unit and the post-process of anchor boxes. The parameters of three CBL units are listed in Table 2. CBL represents a complete convolutional layer, including three operations: Convolution Operation (Conv), Batch Normalization (BN), and Leaky Rectified linear unit (Leak Relu) activation function.

### 2.3. The Siamese CNN with the Cross Stage Partial

The Siamese CNN with the Cross Stage Partial (SiamCSP) neural network is established based on Siamese Net and CSP Net to identify whether the pedestrian is lost or occluded as shown in Figure 7. The SiamCSP is composed of the Feature Enhancement Network, Feature Extraction Layer, and Decision Layer. The parameters of the SiamCSP are listed in Table 3.

(1)Feature Enhancement Network

In the feature extraction layer, POSHE [44] feature enhancement is performed on the frame selection area to achieve a better feature extraction on the prediction frame. It is defined in Equation (4)
(4)$$S_{k}^{e}=\frac{1}{n}\overset{n}{\underset{i=1}{\sum }}\overset{k}{\underset{j=0}{\sum }}p_{i}\left(r_{j}\right),$$
where n is the number of pixels in the entire region, $p_{i}\left(r_{j}\right)$ represents the probability of level j in the region x, and x represents the region a,b,c,⋯,i.

(2)Feature Extraction Layer

To further improve the comparing precision between the initial features (/roi_initial) and the predictive features (/roi_predict), CSPNet is also introduced to the feature extraction network which is shown in Figure 8.

(3)Decision Layer

The proposed SiamCSP uses two fully connected layers as the decision layer, and the Euclidean distance is calculated and defined as the feature keyframe similarity.

(4)Loss function

The cross-entropy classification loss is employed in the training stage. We denote the input of a pair of images as $\left(image_{i},image_{j}\right)$ and set up a new parameter $y_{ij}$. Let $y_{ij}=1$, if $image_{i}$ and $image_{j}$ are of the same person, otherwise $y_{ij}=0$. Then the contrastive loss is adopted for training, as defined in Equation (5)
(5)$$L_{Siam}=\sum_{i,j}y_{ij}\rho_{Siam}^{2}+\left(1-y_{ij}\right)\max{\left(margin-\rho_{Siam},0\right)}^{2}$$
where $\rho_{Siam}^{2}=ǁ\delta \left(image_{i}\right)-\delta \left(image_{j}\right)ǁ$. δ denotes the feature representation by the Siamese CNN, and the margin denotes the desired minimal distance between two images.

### 2.4. Improved Loss Function of the Siam-DFKCF Model

To better ensure the training accuracy, we introduce two additional loss functions into the original YOLO loss function $L_{yolo}$, the similarity loss $L_{Siam}$ calculated by SiamCSP and the loss of the tracking box in the KCF. The $L_{KCF}$ is defined in Equation (8). The global loss function is shown in Equation (6). In the YOLO loss function, the complete intersection over union (CIoU) [45] is adopted to replace the mean square error in the current research, which is defined in Equation (7).
(6)$$Loss=L_{yolo}+L_{Siam}+L_{KCF},$$
(7)$$L_{CIoU}=1-IoU+\frac{\rho_{CIoU}^{2}\left(b_{PB},b_{GT}\right)}{\chi^{2}}\alpha \nu ,$$
(8)$$L_{KCF}=\frac{1}{m}\sum_{i=1}^{m}\left[\max\left(s-\bar{s},margin\right)+\max\left(\rho_{KCF}^{2},margin\right)-2margin\right]$$
where GT and PB denote the ground truth and predict box in Equation (6), $\nu =\frac{4}{\pi^{2}}\left(\arctan{\left(\frac{w}{h}\right)}_{GT}-\arctan{\left(\frac{w}{h}\right)}_{PB}\right)$ in Equation (7), $\left(w,h\right)\;$ is the width and height of the box, and $\rho_{CIoU}^{2}$ is the Euclidean distance between the center point of the GT and PB. χ is the diagonal distance of the smallest closure area that contains both the GT and PB. In Equation (7), $s=Size\left(Current\right)$ and $\bar{s}=Size\left(initial\right)$ denote the size of the KCF tracking box and initial box, and $\rho_{KCF}=\sqrt{{\left(x_{cur}-x_{int}\right)}^{2}+{\left(y_{cur}-y_{int}\right)}^{2}}$ is the Euclidean distance between the center point of the KCF predicted box and the initial box.

## 3. Experiments and Results

### 3.1. Dummy Tracking Experiments Result

#### 3.1.1. Dataset

In our experiment, a dataset was built using a dummy head as an alternative way. A series of pictures were taken at a series of shooting angles, scenes, and light environments. As a result, a total number of 3266 images of the target were obtained. Then, the dummy’s position on the images was labeled using the Label-Image software. 1754 images were randomly selected as the training set, and the other 1502 images were designated as the testing set.

#### 3.1.2. Experiment Arrangement

During the test, a dummy head was arranged to occlude a moving dummy head to compare the Siam-DFKCF and the original KCF algorithms. The experimental environment of this paper was NVIDIA 2070Super, I5-9600 K, and 16 G memory. An MIT motor was used to rotate the dummy head while a stepper motor was used to translate the platform along a guide rail. All of the motors were controlled by an STM32F429 board. The dummy head was used to model a pedestrian as shown in Figure 9. To verify the developed KCF and Siam-DFKCF models for detection and tracking of the moving dummy, experiments were designed and conducted. Moving speed, angular velocity, and tracking distance were studied in the current research. For comparison, the same dataset was used to train models in the Siam-DFKCF and the original KCF algorithms. Thereafter, the tracking tests were performed under the same moving parameters as listed in Table 4.

#### 3.1.3. Experimental Analysis

Anti-occlusion performance of the algorithm

The tracking accuracy of each frame received from the *Usb_Camera* node was recorded to study the anti-occlusion performance of KCF and Siam-DFKCF algorithms. KCF could not preserve features for a long time, and HOG features were shallow; thus, tracking loss usually appeared for the KCF algorithm when suffering a long-term occlusion. This tracking loss was avoided by the Siam-DFKCF algorithm since it could extract and retain the deep features of the missing dummy images. It was irrefutably proven by observations during the test, as compared in Figure 10a,b. The linear speed was arranged to be 1 m/s, the angular speed was π/8 rad/s, and the tracking distance was 5 m. Dummy head A was not occluded until 3.51 s, at which point dummy head B appeared in front of dummy head A until 3.57 s, resulting in the occlusion problem. It was shown that dummy head A was tracked and marked with a blue bounding box in the two algorithms before 3.51 s. However, tracking loss was observed for both the KCF and the Siam-DFKCF algorithms when dummy head A was completely occluded. Since then, the target was lost for the KCF algorithm. On the other hand, dummy head A was retracted for the Siam-DFKCF algorithm. It is estimated that the deep feature of the tracking target in the Siam-DFKCF algorithm enabled the retraction, which is useful for real-time tracking.

During tracking, the camera was controlled to focus on dummy head A. The yaw angular velocity of the camera changed as dummy head A moved from the left side to the right side, which is compared in Figure 11. The $X_{error}$ and $Y_{error}$ represent the relative location of the moving target position from the image center. They were recorded as horizontal deviation and vertical deviation Y in the upper right corner of Figure 10. The missing tracking could be recognized using the recorded yaw angular velocity. Six tests were arranged to study the tracking accuracy of the two algorithms with 0 rad/s angular speed of dummy head A. When an occlusion occurred, the blue box was lost on dummy head B for the two algorithms, resulting in a decrease of the yaw angular velocity between 3.51 s and 4.11 s as shown in Figure 11a. The yaw angular velocity was found to increase to a normal value for the Siam-DFKCF algorithm, which meant a re-tracking was successfully performed. On the contrary, the yaw angular velocity was found to decrease to 0 for the KCF algorithm. Another six tests were arranged to study the angular speed effect of dummy head A as shown in Figure 11b. The maximum angular speeds of 0.26 rad/s and 0.35 rad/s were observed for the non-rotating and π/8 rad/s rotating cases, resulting in a 34.62% increment. It is estimated that occlusion led to a decrease in the angular speed, followed by an abrupt enhancement of the angular speed between 3.51 s and 4.11 s.

Scale adaptation performance of the algorithm

The size of the bounding box was further compared to study the scale adaptation performances of the KCF and Siam-DFKCF algorithms. In the original KCF algorithm, the scale of the extracted image was always the pixel size of the initial target image tracking area; therefore, the relative scale of the target in the image changed according to the relative distance between the camera and the tracking target. If the tracking target moved close or away from the camera, the size of the bounding box should be adaptively changed to capture the exact feature. However, if the size of the bounding box did not change accordingly, the extracted features would be incomplete, or variable background information would be introduced, which might lead to tracking failure.

The width sizes of the tracking box (/roi_predict) of the six cases were compared as illustrated in Figure 12a,b. It was shown that the width of the tracking box (/roi_predict) increased monotonously against velocity in the Siam-DFKCF algorithm. The size of the tracking box gradually increased to a maximum value in the KCF algorithm as shown in Figure 12b. On the other hand, when the similarity of the feature frame was lower than the threshold, the original tracking box would be replaced by the detection box of YOLO-ECA to ensure a modification of the frame size in real-time. Thus, the tracking box was updated and corrected more frequently in the Siam-DFKCF algorithm. It is estimated that the developed Siam-DFKCF algorithm is more suitable for the problem of tracking losses.

To quantitatively analyze the scale adaptation performances of the two algorithms, the subtraction between the initial tracking box size and the final tracking box size is further discussed as shown in Table 5.

The sizes of the initial tracking box and the final tracking box were found to be almost the same for both the KCF and Siam-DFKCF algorithms under 0 rad/s angular velocity. Detailly, the relative deviations were found to be 1.72%, 4.55%, and 5.56% at the speed of 1 m/s, 2 m/s, and 3 m/s respectively in the Siam-DFKCF algorithm. Similarly, the relative deviations were found to be 3.33%, 4.76%, and 6.41% for the KCF algorithm. On the other hand, when the angular velocity increased to π/8 rad/s, the size deviations were abruptly increased to 20.99%, 28.57%, and 38.75% in the KCF algorithm. On the contrary, the size deviations of the Siam-DFKCF algorithm were reduced to 1.52%, 2.6%, and 4.44%. It is implied that better scale adaptation performances were achieved for the Siam-DFKCF algorithm.

Loss and re-tracking of Targets

The similarity was calculated in the decision layer as discussed in Section 2.3. It was found that the similarity of the keyframes began to decrease from 0 s. It is estimated that an occlusion appeared since dummy A was occluded by dummy B, as shown in Figure 13. The similarity was related to the non-occluded part of dummy A, thus the similarity value gradually decreased when the non-occluded part was shrunk step by step. On the other hand, the similarity value gradually increased when dummy A appeared as time goes by. Thereafter, the similarity kept decreasing to a minimum value of 0.11 until 4.07 s in the KCF algorithm since the KCF algorithm did not have a function to retrieve the target. On the other hand, the similarity began to increase to a maximum value of 0.99 until 4.50 s in the Siam-DFKCF algorithm since the Siam-DFKCF algorithm could re-track the target and restore the similarity of the keyframe. It should be noted that a similarity threshold of 0.7 was used empirically in the current research to update the tracking box in the algorithm since 3.42 s. Therefore, the 0.42 s delay was observed. When dummy A rotated with an angular velocity of π/8 rad/s, the average similarity of the re-tracking keyframes could only reach a maximum of 0.89 which was less than the 0.99 of the non-rotated case. It is estimated that the threshold should be reduced adaptively to avoid mis-tracking when the moving target has a large angular velocity.

### 3.2. Pedestrian Tracking Experiments Result

#### 3.2.1. Dataset

In our experiment, a real-time video taken from a school was utilized. There were 121 frames in total and the frame size was 1920 × 1080. The experimental environment of this paper was run on an Inter i5-9600K CPU at 4.30 GHz, Nvidia 2070 Super GPU, and 16 G memory.

#### 3.2.2. Evaluation Criterion

We utilized the evaluation tool provided by VOT [46].

(1)Distance Precision (DP):

The Center Location Error (CLE) between the Tracking Box (TB) and the Ground Truth (GT) was taken into account for calculating the error of location. The CLE was calculated by
(9)$$CLE=\sqrt{{\left(x_{TB}-x_{GT}\right)}^{2}+{\left(y_{TB}-y_{GT}\right)}^{2}},$$

The Distance Precision (DP) was calculated by
(10)$$DP=Num(CLE<\lambda_{DP})/Num\left(total\right),$$
where the $Num(CLE<\lambda_{DP})$ meant the number of frames when the CLE value was less than the threshold $\lambda_{DP}$, and $Num\left(total\right)$ meant the number of frames in the whole data set.

(2)Overlap Precision (OP)

The overlap between the Tracking box (TB) and the ground truth (GT) was taken into account for calculating success scores, and the Intersection over Union (IoU) ratio was calculated by
(11)$$IoU=\left(area\left|TB\cap GT\right|\right)/\left(area\left|TB\cup GT\right|\right)>\lambda \_OP$$

When the IoU value was less than 50%, the position status was defined as false (false positive–FP). In contrast, it was defined as true (true positive–TP) while the method could not produce a position (false negative–FN). This process was repeated by the number of frames in the whole data set, and the ratio of successful frames to the whole frame was found and achieved as Overlap Precision (OP), which was calculated by
(12)$$OP=TF/\left(FP+TF+FN\right)$$

#### 3.2.3. Experiment Results

A speed of 93 FPS (Frames Per Second) was achieved during tracking, and it was easy to track the target face because only one moving face existed in a clean scene as shown in Figure 14a. To verify the tracking accuracy under high moving speed, the person moved fast to show that the target face was locked all the time, as shown in Figure 14b,c. The face was recognized during the whole tracking test even though it was turned around, as shown in Figure 14d,e. Furthermore, it was also recognized when partially covered, as shown in Figure 14f. As a result, we evaluated our algorithm against various special scenarios such as face occlusion, face fast-moving, and face turned. It is shown that the proposed algorithm has strong robustness. Therefore, the real-time face-tracking accuracy was irrefutably validated using the previous tracking experiments. We expect that the proposed algorithm can also be applied in other tacking areas, such as object behavior analysis, anomaly behaviors detection, etc.

The performance of the proposed method was compared with the KCF, Tracking Learning Detection (TLD) [47], Siamese Region Proposal Network (SiamRPN) [48], and Distractor-aware Siamese Region Proposal Network algorithms (DaSiamRPN) [49]. As shown in Figure 15, all algorithms performed well under the slow-moving environment from the 1st frame to the 11th frame. However, the TLD and the KCF algorithm had poor performance in pedestrian tracking which was fast-moving from the 11th frame to the 66th frame. When the face was turned from the 66th frame to the 99th frame, the KCF tracking failed due to the fixed scale, and the TLD was low performing due to the large Tracking Box from the 66th frame. The performance of the SiamRPN and DaSiamRPN was unsatisfactory while the face was occluded from the 99th frame to the final frame. On the contrary, the proposed method tracked the face well from the first frame to the final frame. The Overlap Precision and Distance Precision of the methods are shown in Table 6.

It is shown that the highest DP of 0.934 and the highest OP of 0.909 was achieved using the proposed method in our dataset study; meanwhile, CPU usage and GPU usage were balanced. The TLD, and KCF were traditional filtering algorithms; they had high CPU usage, while the GPU usage was almost zero. On the contrary, the SiamRPN and DASiamRPN algorithm had excellent performance during tracking, and they had a high GPU occupancy for they were deep learning algorithms. As a modification, the method we proposed combined the accuracy of the deep learning algorithm and the low GPU occupancy of the filtering algorithm, and a good balance was achieved between CPU and GPU computations. Overall, the face tracking algorithm designed in this paper had good adaptability in dealing with face tracking.

## 4. Conclusions

A lightweight algorithm with high efficiency is required for an unmanned operating system. Thus, we proposed a deep feature KCF method based on YOLO-ECA and SiamCSP. In the method, the Efficient Channel Attention block (ECA) was introduced into the feature pyramid network (FPN) of YOLO to allocate resources more adaptively. The model focused on the information that was more critical to the current task, and more powerful appearance features were compared to judge similarities for the kernel correlation filter. In this way, the defect in feature association was solved and the tracking-by-detection framework was better used. Additionally, a fire unit was used to make the backbone feature extraction network of the YOLO more lightweight. The tracking efficiency and detecting accuracy were improved. Finally, two series of experiments on the proposed method demonstrate the effectiveness of our algorithm compared to traditional visual tracking methods. In the dummy tracking experiments, influences of moving speed, rotating speed, and occlusion on tracking accuracy and efficiency were studied. It is found that the proposed method had better performance in dealing with anti-occlusion, scale adaptation, and re-tracking problems. Thereafter, the method was used in real-time pedestrian tracking. It is found that the best DP (0.934) and OP (0.909) had been achieved using the proposed method.

In general, the proposed method has good performance in the aspect of anti-occlusion and re-tracking. This lightweight method can significantly improve the high accuracy while maintaining high efficiency. In the future, multiple face tracking will be investigated and discussed in our future studies. Then, we will focus on a similarity representation method of small objects with less computational burden. Finally, we expect that the proposed method can be applied in the area of airborne UAV systems, which will be discussed in future work.

## Figures and Tables

**Figure 1 sensors-23-00482-f001:**
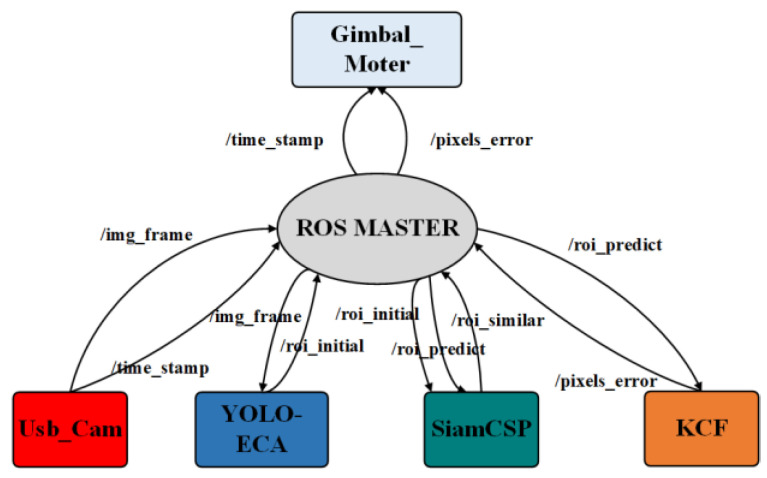
Schematic diagram of ROS nodes of the Siam-DFKCF method.

**Figure 2 sensors-23-00482-f002:**
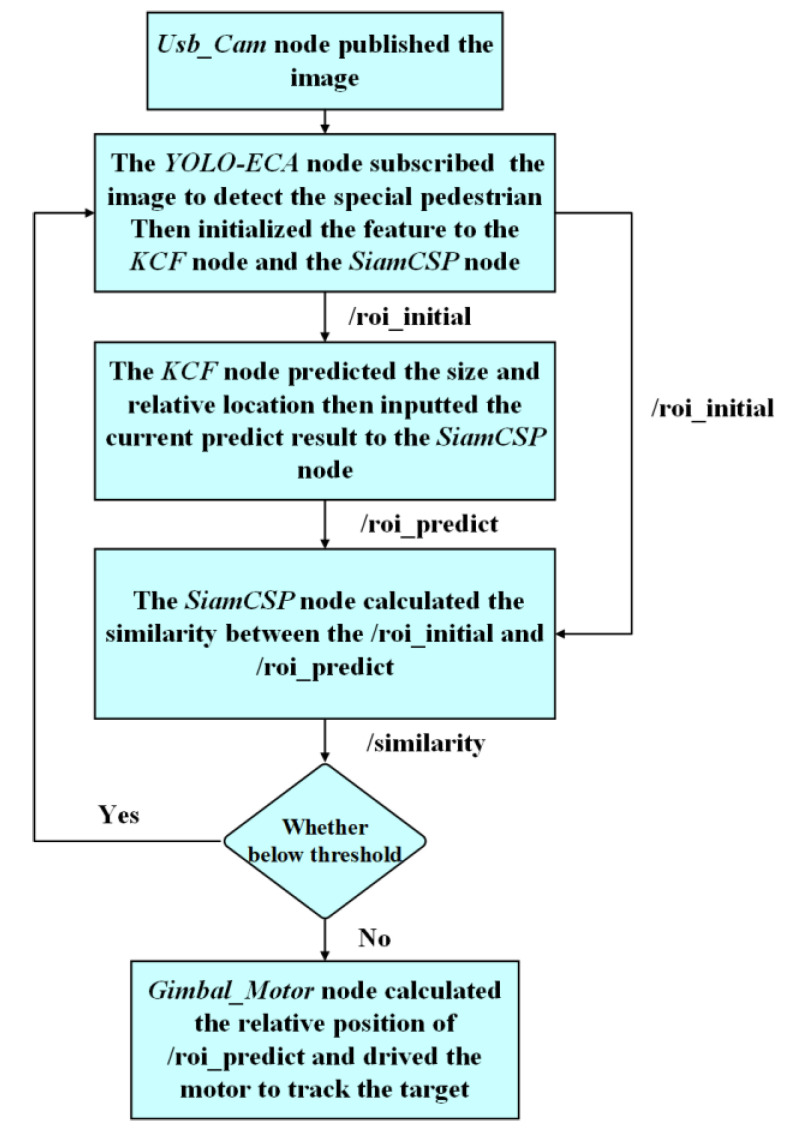
The schematic diagram of the Siam-DFKCF method.

**Figure 3 sensors-23-00482-f003:**
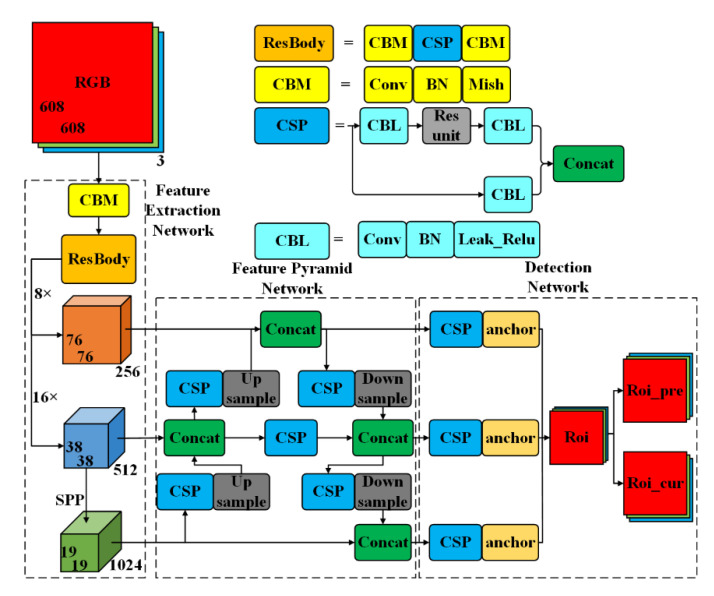
The structure of improved YOLOv3 based on the Efficient Channel Attention (YOLO-ECA).

**Figure 4 sensors-23-00482-f004:**
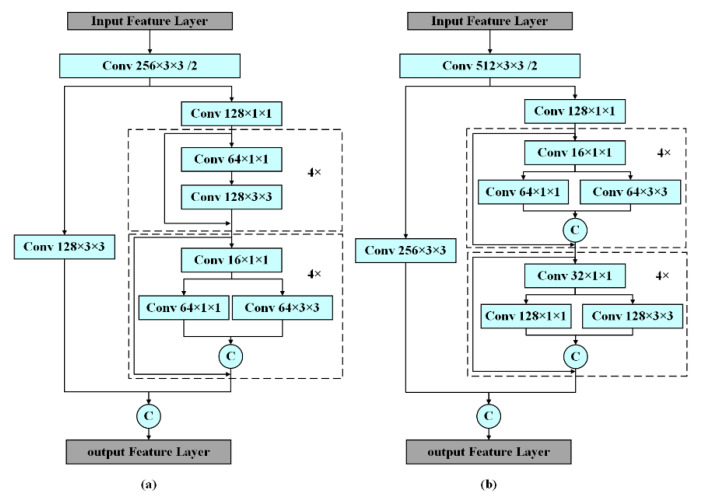
The structure of Squeeze Cross Stage Partial Body (SCSPBody) ((**a**) SCSPBody1, (**b**) SCSPBody2).

**Figure 5 sensors-23-00482-f005:**
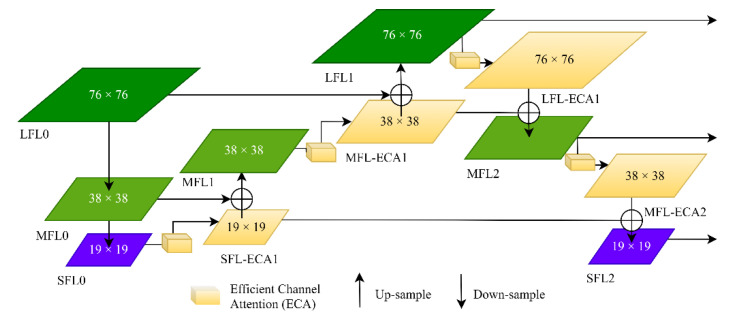
The structure of the Feature Pyramid Network.

**Figure 6 sensors-23-00482-f006:**
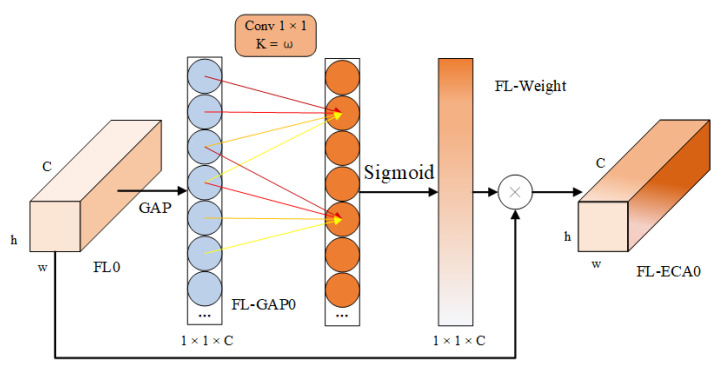
The structure of the Efficient Channel Attention block (ECA).

**Figure 7 sensors-23-00482-f007:**
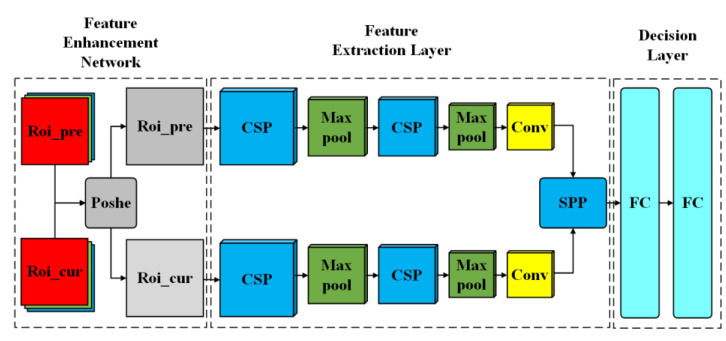
The structure of the Siamese CNN with Cross Stage Partial (SiamCSP).

**Figure 8 sensors-23-00482-f008:**
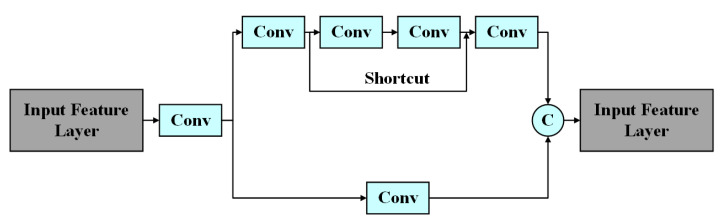
The structure of the Cross Stage Partial (CSP)block.

**Figure 9 sensors-23-00482-f009:**
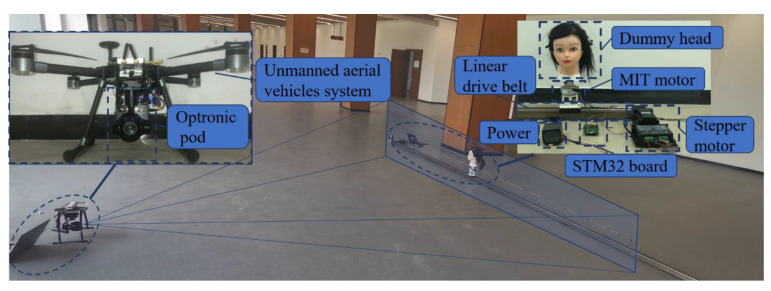
Schematic diagram of experimental equipment.

**Figure 10 sensors-23-00482-f010:**
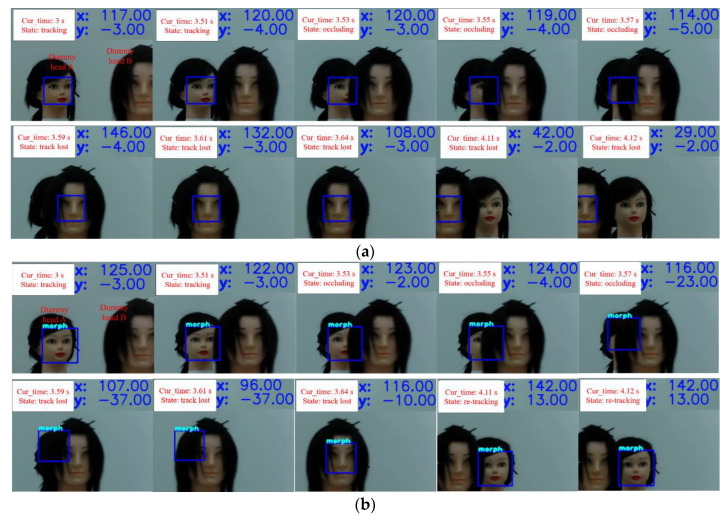
The tracking process of the dummy ((**a**) the KCF algorithm, (**b**) the Siam-DFKCF algorithm).

**Figure 11 sensors-23-00482-f011:**
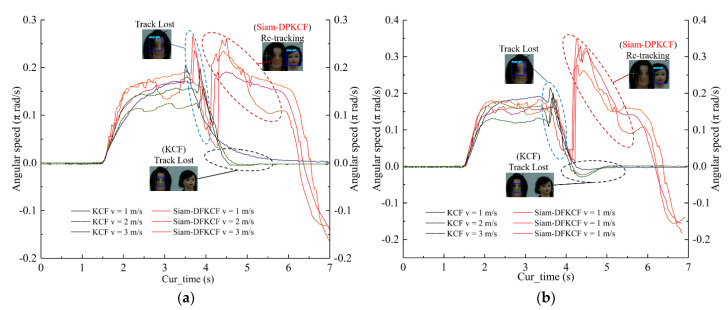
Angle speed of the camera between the Siam-DFKCF and KCF ((**a**) angular speed of 0 rad/s, (**b**) angular speed of π/8 rad/s).

**Figure 12 sensors-23-00482-f012:**
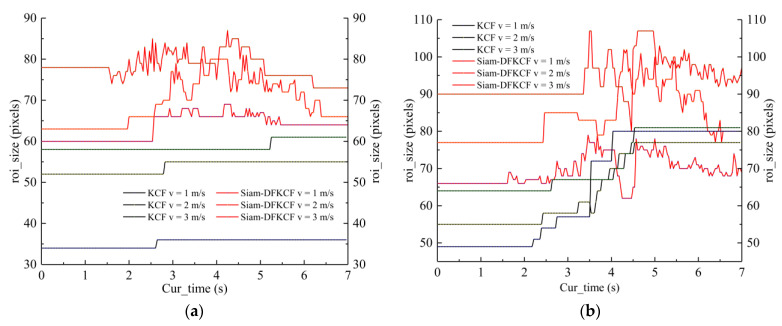
The real-time size of the tracking box ((**a**) angular speed of 0 rad/s, (**b**) angular speed of π/8 rad/s).

**Figure 13 sensors-23-00482-f013:**
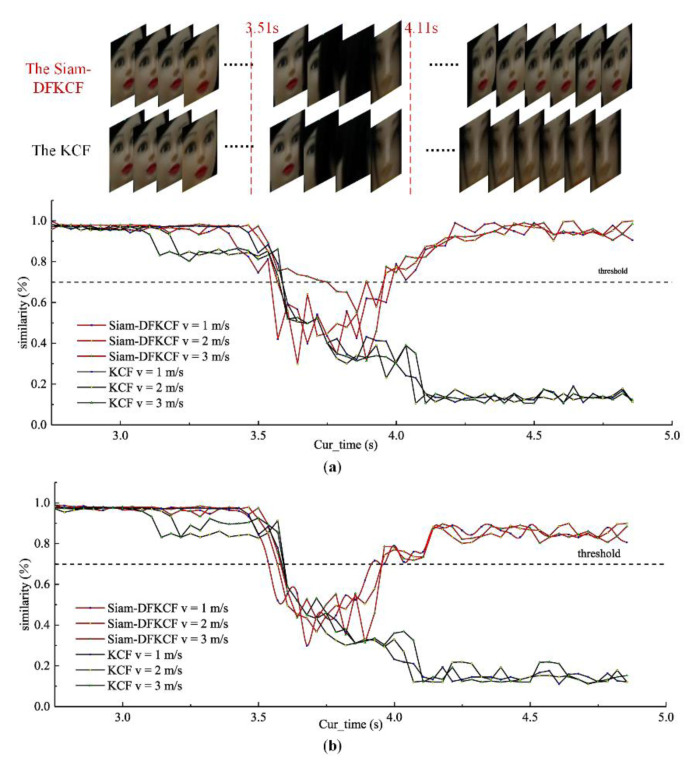
The similarity of the keyframes ((**a**) angular speed of 0 rad/s, (**b**) angular speed of π/8 rad/s).

**Figure 14 sensors-23-00482-f014:**
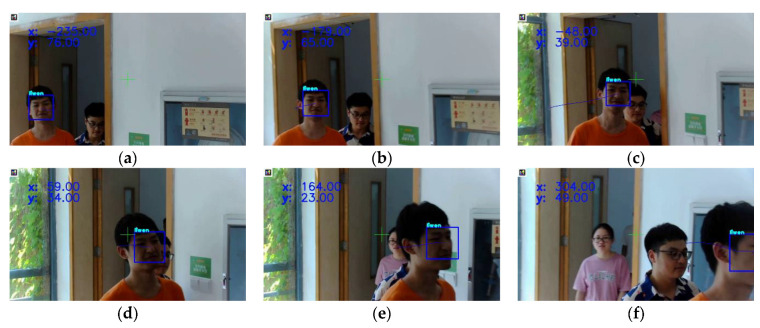
Experimental results of tracking faces ((**a**). the 11th frame of the test video, (**b**). the 32th frame of the test video, (**c**). the 51th frame of the test video, (**d**). the 79th frame of the test video, (**e**). the 92th frame of the test video, (**f**). the 118th frame of the test video).

**Figure 15 sensors-23-00482-f015:**
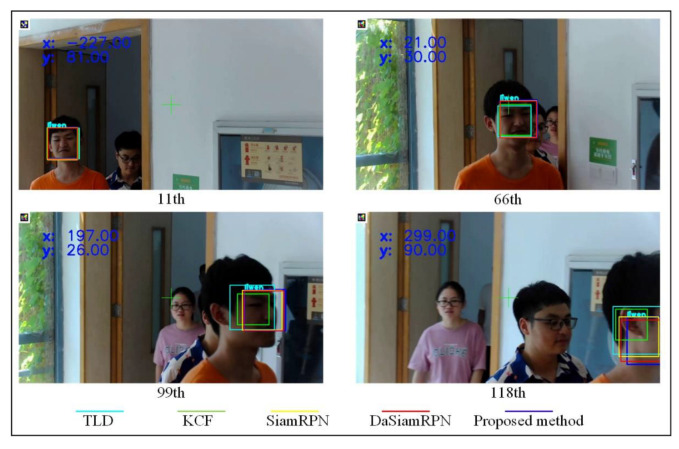
Results of different tracking algorithms.

**Table 1 sensors-23-00482-t001:** The Feature Extraction Network of YOLO-ECA.

		Type	Filters	Size	Output
		Conv	32	3 × 3/2	32 × 304 × 304
		Conv	64	3 × 3/1	64 × 304 × 304
		Conv	64	3 × 3/2	64 × 152 × 152
		Conv	128	3 × 3/1	128 × 152 × 152
Conv 128 × 3 × 3		Conv	256	3 × 3/2	256 × 76 × 76
		Conv	128	1 × 1/1	
	4×	Conv	64	1 × 1/1	
		Conv	128	3 × 3/1	
		Residual			
	4×	Conv	16	1 × 1/1	
		Conv	64	1 × 1/1	
		Conv	64	3 × 3/1	
		Fire			
Conv 256 × 3 × 3		Conv	512	3 × 3/2	512 × 38 × 38
		Conv	128	1 × 1/1	
	4×	Conv	16	1 × 1/1	
		Conv	64	1 × 1/1	
		Conv	64	3 × 3/1	
		Fire			
	4×	Conv	32	1 × 1/1	
		Conv	128	1 × 1/1	
		Conv	128	3 × 3/1	
		Fire			
	1×	Conv	512	3 × 3/2	512 × 19 × 19
	Direct	Conv Max5 × 5	Conv Max9 × 9	Conv Max13 × 13	1024 × 19 × 19
	SPP				

**Table 2 sensors-23-00482-t002:** The Detection Network of YOLO-ECA.

		Type	Filters	Size	Output
Large	3×	Convolutional	256	3 × 3/1	256 × 76 × 76
		BN			
		LeakReLU			
Medium	3×	Convolutional	512	3 × 3/1	512 × 38 × 38
		BN			
		LeakReLU			
Small	3×	Convolutional	1024	3 × 3/1	1024 × 19 × 19
		BN			
		LeakReLU			

**Table 3 sensors-23-00482-t003:** The parameters of the SiamCSP.

		Type	Filters	Size	Output
		Resize			3 × 304 × 304
		Poshe			3 × 304 × 304
Conv 64 × 3 × 3		Conv	128	3 × 3/2	128 × 152 × 152
		Conv	64	3 × 3/1	
	1×	Conv	32	1 × 1/1	
		Conv	64	3 × 3/1	
		Residual			
		Conv	64	3 × 3/1	
		MaxPooling		2 × 2	128 × 76 × 76
Conv 128 × 3 × 3		Conv	128	3 × 3	256 × 38 × 38
		Conv	64	3 × 3/1	
	1×	Conv	32	1 × 1/1	
		Conv	64	3 × 3/1	
		Residual			
		Conv	128	3 × 3/1	
		MaxPooling		2 × 2/1	256 × 19 × 19
		Conv	512	3 × 3	512 × 19 × 19
	Direct	Conv Max5 × 5	Conv Max9 × 9	Conv Max13 × 13	512 × 7×7
		SPP			
		FC			
		FC&Sigmoid			Similarity

**Table 4 sensors-23-00482-t004:** The motion parameters for comparison experiments.

Number	Tracking Distance (m)	Linear Velocity (m/s)	Angular Velocity (rad/s)
1	5	1	0
2	5	1	π/8
3	5	2	0
4	5	2	π/8
5	5	3	0
6	5	3	π/8

**Table 5 sensors-23-00482-t005:** The results of the tracking box of different algorithms.

Detection Method	The Difference in Sizes Between the Initial and Final Tracking Box					
** $\mathrm{linear}\;\mathrm{velocity}\left(m/s\right)$ **	1		2		3	
** $\mathrm{Angular}\;\mathrm{velocity}\left(rad/s\right)$ **	0	π/8	0	π/8	0	π/8
**Siam-DFKCF**	1.72%	1.52%	4.55%	2.6%	5.56%	4.44%
**Traditional KCF**	3.33%	20.99%	4.76%	28.57%	6.41%	38.75%

**Table 6 sensors-23-00482-t006:** The results of the tracking box of different algorithms.

Method	DP	OP	FPS	GPU Cost	CPU Cost
**TLD**	0.653	0.628	32	0	36
**KCF**	0.694	0.645	241	0	33
**SiamRPN**	0.868	0.843	134	32	0
**DaSiamRPN**	0.917	0.892	117	34	0
**Proposed method**	0.934	0.909	93	23	18

**FPS:** Frames Per Second.

## Data Availability

The data that support the findings of this study are available from the corresponding author upon reasonable request.

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
