# Peer review of "Siam Deep Feature KCF Method and Experimental Study for Pedestrian Tracking"

_sensors, 2023, doi:10.3390/s23010482_

Round 1

Reviewer 1 Report

Pedestrian tracking using Siam deep feature is proposed in this manuscript. The manuscript is interesting and well-written. However, the following are my concerns in this manuscript. 

Abstract:

The abbreviations such as YOLO, ECA, KCF, and DFKCF are not expanded in the abstract section. 

Abstract should be rewritten, currently, it lacks focus and objective. What is siamese CNN and its advantages, what is SiamCSP, and a few other unanswered and undefined techniques are mentioned.

Which version of the YOLO is used? It should be mentioned in the abstract

Authors claim that their model is providing solutions for target tracking but to what extent and what the improvement in % compared with benchmark techniques should be mentioned. 

Introduction:

Again, in this section also few abbreviations found without expansions. 

Line No. 40 - Instead of starting with "The authors in [9]" the last name of the authors can be mentioned. For eg., "Andrew et al [9] show"

Introduction section should introduce the background and techniques used for this research. But I feel the introduction and background of neural networks, CNN, siamese networks, and YOLO models are missing. State-of-the-literature review is meaningless without the basics. 

Clearly define the research questions/problem statements identified from the literature survey

Summary of the contributions should be included. 

Section-wise outline should be included. 

Materials and Methods:

In section 2.2, user presented the improved YOLOV3 model. It is essential to discuss the original YOLOV3 model then the improvements brought to the original network should be highlighted. Techical presentation is required rather than experimental details. Experimental details can be added in other sections. 

What is Mish activation function in Line No. 143

Training and test accuracy of the proposed model should be reported.

Which dataset is used for training the model? As of now it is mentioned that a dataset of 121 frames is used. Which is very less for a deep learning model. Provide details on the training and test set. 

As per the results, I understood that only one face is tracked at a time. Is it right? If so, why not multiple face tracking is implemented. 

Discussion section to discuss the limitation and merits of the proposed model can be included. 

Future work in conclusion should be enhanced. 

Reviewer 2 Report

Authors have designed a lightweight network with fast-running abilities to track a moving body even though it may be occluded using  Siam Deep Feature KCF method. It appears they have achieved good results but overall design of proposed network is very difficult to follow from presented information.

 Authors are advised to thoroughly proofread the paper and re-write the paper for better understanding in terms of usage of language. For example Authors have written P2 ln 66,67

“Detailly, in the part of data association, due to the limited computing performance of airborne UAVs, the deep tracking method is difficult to be applied to ensure real-time  tracking.”

Intent of authors is not clear.

Another example in Ln 77 authors use acronym CSK which is not explained.

Complete framework is not adequately explained from figure 1.

I was unable to find figure 4, 4a, 4b.

Seriously wished to read this manuscript having proofread.

Readability of manuscript needs to be improved before it is subjected to further review.

Round 2

Reviewer 1 Report

I appreciate the author's efforts in addressing my concerns. I am happy with the revised version. 

Author Response

Dear reviewer, thanks for your most valuable suggestions, we have proofread the acronym in the manuscript

Reviewer 2 Report

This is an interesting study  and the authors have created a unique dataset. The paper has presented a robust method for pedestrial tracking in case of occlusion. It may be useful for the researchers working in the field of pedestrian tracking.  However as mentioned in my previous review also authors should meticulously read each sentence to  ensure proper spelling punctuation and grammer.

For example

Abstract page 1 line 13 

"a lightweight YOLOv3-based (you only look once version 3) mode which had Efficient Channel Attention (ECA) was integrated into the CFs algorithm 14 to provide deep features."

This kind of error is not expected at this stage of review. The authors should use professional academic proofreading service.

Author Response

Dear reviewer, thanks for your most valuable suggestions, we have proofread the acronym in the manuscript, and the manuscript has been polished thoroughly. We expect our manuscript meets the requirements of our journal.